# The physical profile of female cricketers: An investigation between playing standard and position

**Thomas A. Brazier**  [1,2☯]*, **Jamie Tallent**[3,4☯], **Stephen D. Patterson**[2], **Louis P. Howe**[3], **Samuel J. Callaghan** [2☯]

1 Department of Sport Science and Medicine, Northamptonshire County Cricket Club, Northampton, United Kingdom, 2 Centre for Applied Performance Sciences, St Mary's University, Twickenham, United Kingdom, 3 School of Sport, Exercise and Rehabilitation Sciences, University of Essex, Essex, United Kingdom, 4 Department of Physiotherapy, Faculty of Medicine, Nursing and Health Science, School of Primary and Allied Health Care, Monash University, Victoria, Australia

☯ These authors contributed equally to this work.
* tombrazier96@gmail.com

## Abstract

The primary aim of this study was to present the physical profile of female cricketers. Secondary, was to assess any differences between playing standard (professional vs. non-professional) and position (seam bowler vs. non-seam bowler). Fifty-four female cricketers (professional seam bowler [n = 16]; professional non-seam bowler [n = 17]; non-professional seam bowler [n = 10]; non-professional non-seam bowler [n = 11]) undertook a battery of physical and anthropometric assessments during the off-season period. Participant's physical profile was assessed via the broad jump, countermovement jump, isometric mid-thigh pull (IMTP), 20 m sprint, run-2 cricket specific speed test, and Yo-Yo Intermittent Recovery Test Level-1 (Yo-Yo-IR1). The sum-of-eight skinfold measurement was also recorded for professional cricketers only. Differences between playing standard and position were assessed with a two-way ANOVA. Seam bowlers possessed a significantly ($p < 0.04$) greater stature and had a higher body mass than non-seam bowlers. Non-seam bowlers recorded significantly ($p < 0.01$) further broad jump, higher normalised peak vertical force during the IMTP, and ran greater distances during the Yo-Yo-IR1. Professional cricketers produced significantly further run distances for the Yo-Yo-IR1 and faster run-2 times for the dominant turning side than non-professional cricketers. This study provides valuable insights into the physical profile of female cricketers across playing standards and positions which practitioners can use to direct and enhance training outcomes.

## Introduction

Cricket is a bat and ball game played between two teams of eleven players. Cricket match-play is characterised by extended periods of low intensity activity, interspersed with bouts of high-intensity actions, such as bowling, running between the wickets or fielding the ball [1, 2]. One of the unique characteristics of cricket, is the different match formats which vary in match

**Data Availability Statement:** All relevant data are within the paper and its Supporting information files.

**Funding:** The authors received no specific funding for this work.

**Competing interests:** The authors have declared that no competing interests exist.

duration (i.e., Twenty20 [T20], one-day and multi-day matches). Typically, women's cricket is played over either T20 or one-day formats. Despite the different match formats, the required skills, and physical capacities of players, be that batting, bowling, or fielding remain consistent.

The extent to which a player is successful in cricket relies on a range of technical and physical capabilities. These underpinning physical capacities have been considered important, regardless of playing positions [3, 4]. A seam bowler will perform a linear sprint to the crease to deliver the ball as part of their bowling action, while a batter will need to sprint between the wickets to score a run. Regardless of playing position, improved lower-limb strength and power would be advantageous for performance (e.g., to aid a seam bowler in bracing their front leg during front foot contact of the delivery stride or to increase jump height or speed in the field when attempting a catch or run-out a batter) [5, 6]. Furthermore, enhanced aerobic endurance, has been linked to better performing seam bowlers and batters, with even the shortest match format, T20, lasting approximately 3 hours [7]. However, the importance of these physical capacities to overall individual, and team success does appear to vary between playing positions. For example, Weldon, Clarke [3] reported that male batters demonstrated greater lower-limb power than seam bowlers, as measured via the countermovement jump (CMJ) test. Whether these anticipated consistencies and differences in physical capacities between playing positions are also present among female cricketers has yet to be investigated within the scientific literature.

To date, only one study has reported the physical capacities of female cricketers [8]. This investigation, while providing valuable insights into the physical capacities of female cricketers, was delimited to elite, professional seam bowlers. No research has investigated the physical capacities of non-seam bowlers or non-professional female cricketers. There is a clear need for a better understanding of the physical profile of female cricketers based upon playing standard and position, as to enhance strength and conditioning practices within this population.

Therefore, the aim of this study are to: 1) develop a physical profile of the female cricketer and 2) identify if there are any differences in physical capacities between playing standard and position. Given the lack of research investigating the physical profile of female cricketers, this research will provide unique insights and assist practitioners in identifying optimal profiles and enhancing training practices.

## Materials and methods

All Participants received a clear explanation of the study, including the risks and benefits of participation, and provided written and informed consent prior to participation. The research was approved by the institutional ethics committee, in agreement with the declaration of Helsinki, version seven.

### Participants

Fifty-four female cricketers (age = 23.0 ± 3.90 years; stature = 1.67 ± 0.06 m; body mass = 63.34 ± 7.29 kg) were recruited for this study. This sample was comprised of thirty-three professional female cricketers (age = 24.9 ± 3.10 years; stature = 1.68 ± 0.06 m; body mass = 64.90± 6.71 kg) and twenty-one non-professional female cricketers (age = 22 ± 3.67 years; stature = 1.67 ± 0.05 m; body mass = 60.94 ± 7.67 kg). Players were separated into four groups, based upon their playing standard and position (professional seam bowler [n = 16]; professional non-seam bowler [n = 17]; non-professional seam bowler [n = 10]; non-professional non-seam bowler [n = 11]). Professional cricketers were defined as having a United Kingdom (UK) based, domestic, county team senior contract, whereas non-professional was classified as cricketers who were part of a domestic, county academy pathway and did not

possess a professional senior team contract. The positional classification was defined by the head coach of each respective team, with a seam bowler defined as a player whose primary role within the team was seam bowling. Non-seam bowlers were classified as cricketers that occupied all other positions (i.e., spin bowlers, batters and wicket keepers). The Participant's inclusion criteria required a current domestic county level senior team contract or part of a domestic county cricket academy pathway, ≥18 years of age, and free from any existing medical conditions that were contrary to participation, as determined by a physical activity readiness questionnaire, and discussion and clearance with each domestic teams, qualified physiotherapists. The physical capacity assessments included the following tests: broad jump, CMJ, IMTP, 20 m sprint with time intervals recorded at 0–10 and 0–20 m, run-2 (dominant and non-dominant turns) and the Yo-Yo-IR1. A 3-minute rest period was provided between the performance of each test. All chosen physical tests are currently part of the English and Wales Cricket Board regional profiling assessment. A standardised warm-up, consisting of jogging, dynamic stretching, progressive speed runs and plyometrics drills was used for all Participants. All Participants were instructed to refrain from vigorous physical exercise, caffeine, or any adrenergic substance in the 24-hours prior to data collection, and to maintain their normal dietary habits. Testing was conducted in the indoor training facilities of each of the six domestic regional teams included within the study, at the same time of day.

## Procedures

The study was undertaken during the off-season phase of the cricket competition as part of the players required physical testing. Participants were required to complete a single testing session.

**Anthropometric.** Prior to the commencement of data collection, the participant's chronological age, height and body mass were recorded by the authors of this study, participants were in minimal clothing for anthropometric procedures, and this was standardised across all participants. Stature was measured barefoot using a stadiometer using the stretch height procedure [9] (Ecomed Trading, Seven Hills, Australia). Body mass was recorded using digital scales (Tanita Corporation, Tokyo, Japan).

**Broad jump.** The broad jump assessment was used to indirectly measure horizontal power production. The Participants placed their heels on the starting line, with their feet parallel. With a simultaneous arm swing and crouch, the Participants then leapt forward as far as possible, ensuring a 2-footed landing. Participants had to "stick" the landing for the trial to be counted. If this was not done, the trial was dis-regarded and another attempted. No restrictions were placed on range of motion during the countermovement, or arm swing used. Distance was measured to the nearest 0.01 m using a standard tape measure (HART, Sport, Aspley, Australia) perpendicularly from the front of the start line to the posterior surface of the heel at landing [10, 11]. Participants completed three trials of the broad jump, with a 1-minute rest between trials. The best score of the three trials was used for analysis.

**Counter movement jump.** The CMJ assessment was used as an indirect measure of vertical power production. All CMJs were performed on two portable force plates (Pasco Force Platform PS-2141, Roseville, CA, USA) sampling at 1000 hertz. Briefly, Participants performed three trials with a 1-minute rest between trials. Participants were instructed to jump as high as possible, and no restrictions were placed on the lower-limb countermovement range during the eccentric phase of the jump or arm swing throughout the movement [3, 12]. Customise computer software (ForceDecks, VALD Performance, Newstead, Australia) was utilised to determine jump height. The highest jump recorded for each Participant was utilised for analysis.

**Isometric mid-thigh pull.** The IMTP is a reliable test of lower-limb maximal isometric strength [13]. The procedures used to perform the IMTP were based upon previous research, with the test completed on the aforementioned two force plates within a customised power rack. The customised power rack allowed the bar to be fixed for each Participant. Briefly, Participants were instructed to grip the bar in a position similar to that of a second pull of a power clean, with an upright trunk position so that their shoulders were in line with the bar, in their preferred position for the pull. This positioning allowed participants to maintain their preferred position between a knee angle range of 120˚-150˚, and a hip angle range of 125˚-145˚ during testing, which has been shown within previous research to result in no statistical difference in peak force variables when performing the IMTP [14]. Wrist straps were utilised for all Participants to ensure that grip strength was not a limiting factor for the measurement of lower-limb strength. Participants were instructed to pull as hard as possible on the bar while driving their feet as hard as possible into the force plate for 5-seconds [6, 14, 15]. Each Participant was required to complete two trials of the IMTP, with a 2-minute rest between trials. All participants were familiar with the IMTP testing procedures. The peak vertical force and peak vertical force normalised to body weight across all trials was used for analysis.

**10 and 20—Meter sprint.** Three maximal 20 m sprints with a 3-minutes rest between trials were performed. Sprint time was recorded by a timing lights system (Browser Timing Systems, Draper, USA). Gates were placed at 0 m, 10m and 20 m, at a height of 1.2 m and a width of 1.5 m, to measure time over 0–10 m and 0–20 m intervals. Participants began the sprint from a standing, staggered stance, 50 cm behind the start line to trigger the first gate and were instructed to accelerate from the starting line and sprint through all gates. Participants self-selected the lead leg, which remained constant throughout testing. If the Participant rocked backwards or forwards before starting, the trial was disregarded and repeated [12]. Time for each interval was recorded to the nearest 0.01 seconds, with the fastest trials used for analysis.

**Run-2.** The run-2 test is designed to assess the Participant's ability to run between the wickets within a match and has previously been used within the literature as a cricket specific measure of speed [12]. The run-2 test times the Participant's ability to run between two lines 17.68 m apart (distance between the two creases on a cricket pitch). The timing light gate was placed at the start line/crease and set at a height of 0.6 m. Participants were required to use and complete the test with a cricket bat, but with no other batting equipment (i.e., pads, gloves, or helmet). Participants started in the split stance position, 0.5 m behind the start line with the cricket bat in hand. Participants were instructed to complete the test as quickly as possible, making sure to slide the bat over the crease mark at the turn and start/finish, as they would within a match [12]. Participants were required to complete three trials turning off each the left and right foot in a randomised order. A 3-minute recovery was given between each trial. Total time to complete the run-2 test was recorded to the nearest 0.01 seconds. For analysis the side (i.e., left or right turning foot) with the fastest turn time was defined as the dominant side, with the other turning side labelled the non-dominant side [16]. The fastest trial from both the dominant and non-side turning side was used for analysis.

**YO-YO IR L2- Yo-Yo Intermittent Recovery Test Level 1.** The Yo-Yo-IR1 was used as to assess the Participant's aerobic endurance capacity. The protocol for the Yo-Yo-IR1 has been outlined in previous research [17]. Briefly, the test consists of running between two lines (shuttle) set 20 m apart. A further cone was placed 5 m back from the start/finish line for the Participants to walk to during the 10 second active recovery between shuttles. The increasing speed of each shuttle was controlled by an audio beep. The test ended when a Participant failed to complete two individual shuttles in the required time. The total running distance completed by the Participant was used for data analysis.

**Skinfold measurements.** Skinfold assessment has previously been shown to link to physical capacity metrics and also forms part of the testing procedures of the English and Wales Cricket Board. Skinfold measurements were only able to be collected on the professional female cricketers. The sum-of-8 skinfold thickness was completed by an International Society for the Advancement of Kinanthropometry (ISAK) practitioner. The standardised sum-of-8 skinfold sites (bicep, tricep, subscapular, supraspinale, suprailiac, abdomen, mid-thigh and medial calf) and procedures recommended by ISAK were used. All measurements were recorded using Harpenden callipers (British Indicators, Hertfordshire, United Kingdom). The ISAK method has been shown to be highly reliable [18].

### Statistical analyses

Descriptive statistics (mean ± standard deviation) were used to describe each variable. Normality of data was assessed by visual analysis of Q-Q plots. Levene's test was used to check for homogeneity of variance before analyses. To detect differences between playing standard (professional vs. non-professional) and position (seam bowler vs. non-seam bowler) a 2 x 2 ANOVA was conducted. Alpha level was set at 0.05. If a significant interaction was detected, a pairwise comparison using a Bonferroni post hoc. As only the professional cricketers were assessed for changes in skinfold measurement, a independent samples t test was used to assess any difference between positions among professional cricketers. These statistics were computed using the Statistics Package for Social Sciences Version 24.0 (IBM, Armonk, USA). The standardised magnitude of effect sizes (ES) difference was examined between groups (professional seam bowler, professional non-seam bowler, non-professional seam bowler, non-professional non-seam bowler). The effect magnitude was assessed on the following scale: less than 0.25 was considered a trivial effect; 0.25 to 0.49 a small effect; 0.50 to 1.00 a moderate effect; greater than 1.00 a large effect [19].

## Results

All investigated variables were deemed to be normally distributed as determined by the Q-Q plot analysis. The mean and standard deviation for each assessed anthropometric variable were professional seam bowler (stature = 1.69 ± 0.06 m; body mass = 66.53 ± 6.10 kg), professional non-seam bowler (stature = 1.67 ± 0.06 m; body mass = 63.31 ± 7.06 kg), non-professional seam bowler (stature = 1.69 ± 0.06 m; body mass = 63.48 ± 6.84 kg) and non-professional non-seam bowler (stature = 1.64 ± 0.02 m; body mass = 58.64 ± 7.97 kg). There was a significant difference in stature ($F_{1,50}$ = 4.4; $p$ = 0.04;), and body mass ($F_{1,50}$ = 4.3; $p$ = 0.04) between positions. Seam bowlers were significantly taller and had a greater body mass when compared to non-seam bowlers. There were no other significant main effects or interactions between stature, and body mass.

The individual and mean results across physical performance and skinfold measurements across playing standard and position are presented in Fig 1. Effect size difference in physical performance and anthropometric profiles between playing standard and position are presented in Fig 2. There was a significant ($F_{1,48}$ = 5.0; $p$ = 0.03) difference in broad jump distance, with non-seam bowlers jumping ~9% further than seam bowlers. Non-seam bowlers also recorded a significantly ($F_{1,48}$ = 8.8; $p$ = 0.01) higher normalized IMTP value than seam bowlers. While no difference there was a *large* effect (ES > 1.0) for professional non-seam cricketers to produce higher normalized peak force values during the IMTP. Professional cricketers recorded a difference ($F_{1,48}$ = 13.5; p < 0.01) in higher peak absolute vertical force during the IMTP than non-professional cricketers by ~20%. Professional cricketers were also significantly quicker ($F_{1,45}$ = 4.9; $p$ = 0.03;) than the non-professional cricketers (professional = 6.86 seconds

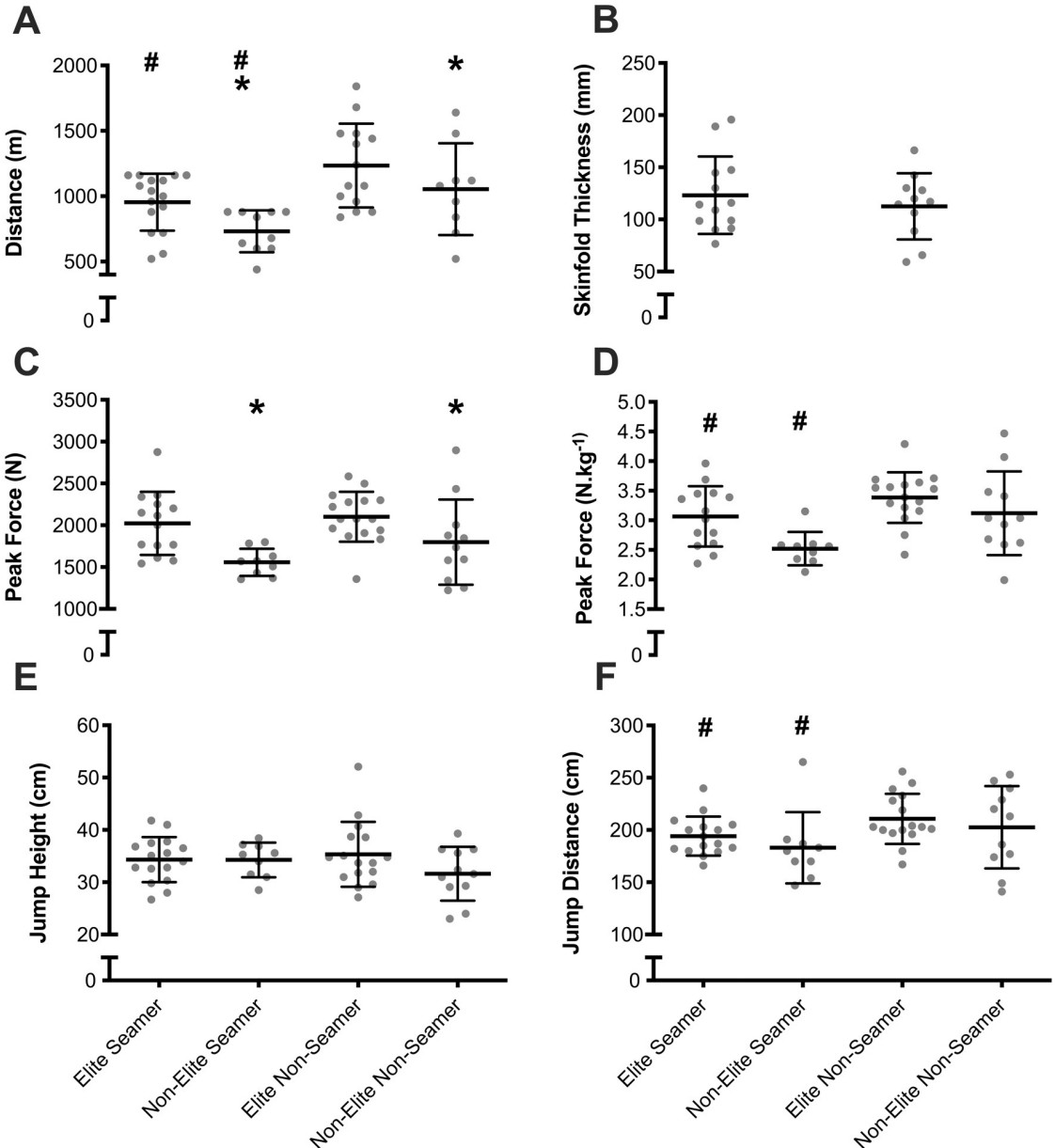

**Fig 1.** The individual and mean Yo-Yo-IR1[19] run distance (A), sum-of-8 skinfold thickness (B), isometric mid-thigh pull (IMTP) absolute peak force (C), IMTP relative peak force (D), countermovement jump height (E), and standing broad jump distance (F), across playing standard and position. #Denotes a significant ($p < 0.05$) main effect difference between seam bowlers and non-seam bowlers; *Denotes a significant ($p < 0.05$) main effect difference between professional and non-professional players.

vs. non-professional = 7.02 seconds) to complete the run-2 on their dominant foot. The Yo-Yo-IR1 test revealed a significant difference in both playing standard ($F_{1,45} = 6.5$; $p = 0.01$;), and position ($F_{1,45} = 14.4$; $p < 0.01$;). Professional cricketers recorded, a *moderate* effect, ~20% further run distance than non-professional cricketers, whereas non-seam bowlers, had a *large* effect, ~29% further run distance than seam bowlers. There was no significant interaction effect ($p > 0.05$) between playing standard and position. However, there was a large effect

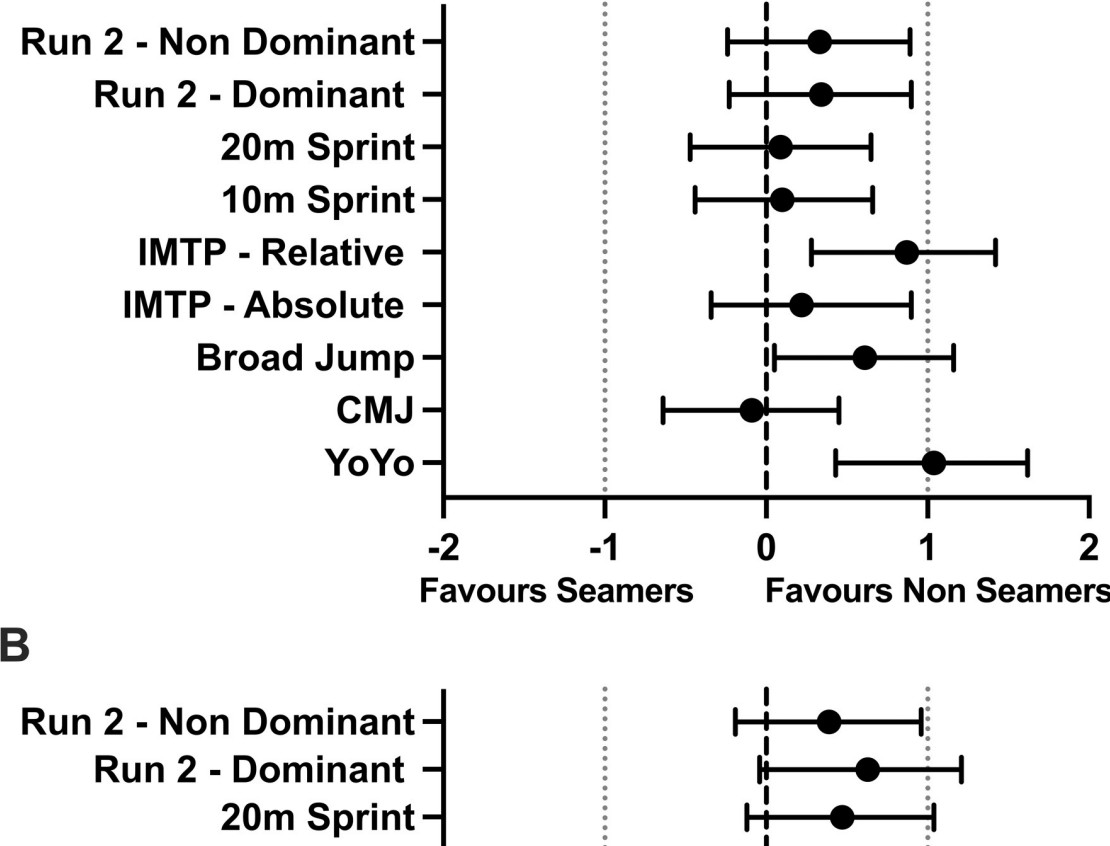

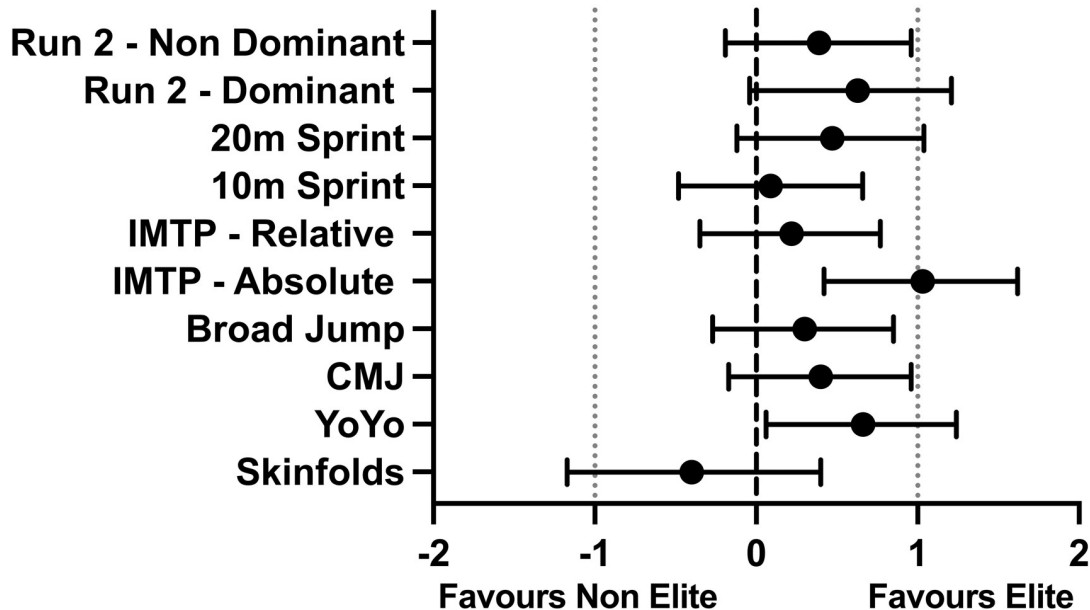

**Fig 2. Effect sizes for physical capacity and anthropometric tests between playing standard and position.** IMTP = isometric mid-thigh pull; CMJ–countermovement jump.

(ES > 1.0) run distance for professional non-seam players in comparison to all other groups. There were no other significant differences ($p > 0.05$) in any physical performance or anthropometric variables. The mean and standard deviation sprint and run-2 test times across playing standard and position are displayed in Table 1.

**Table 1. The mean ± standard deviation sprint and run-2 test times across playing standard and position.** m = metres; s = seconds; N-dominant = non-dominant; n = number of participants.

| | 10 m (s) | 20 m (s) | Run-2 dominant (s) | Run-2 N-dominant (s) |
|---|---|---|---|---|
| Professional seam bowler (n = 15) | 1.98 ± 0.10 | 3.41 ± 0.13 | 6.93 ± 0.20* | 7.05 ± 0.33 |
| Non-professional seam bowler (n = 14) | 1.96 ± 0.06 | 3.43 ± 0.08 | 7.04 ± 0.19 | 7.10 ± 0.20 |
| Professional non-seam bowler (n = 10) | 1.95 ± 0.76 | 3.36 ± 0.13 | 6.79 ± 0.30* | 6.87 ± 0.33 |
| Non-professional non-seam bowler (n = 10) | 1.98 ± 0.12 | 3.47 ± 0.17 | 7.01 ± 0.31 | 7.09 ± 0.40 |

*Denotes a significant ($p < 0.05$) main effect difference between professional and non-professional players.

## Discussion

This is the first study to investigate the physical profile of professional and non-professional female cricketers across playing positions. The main findings of this study demonstrated a trend of non-seam bowlers possessing superior physical performance, with enhanced lower-limb power and strength, and aerobic endurance when compared to seam bowlers [6]. This is somewhat surprising given the proposed physical demands of seam bowling reported within male cricket. The professional female cricketers were also found to possess superior change of direction ability and aerobic endurance performance in comparison to non-professional female cricketers. This may be a result of a greater exposure to structured physical training, resulting in a higher training age, and improved physical performance [20, 21]. Overall, the results of this study provide valuable insights into the physical profile of female cricketers across playing standards and positions which coaches and strength and conditioning practitioners can utilise as normative values and to inform training practices.

Seam bowlers were reported to be taller and have a greater body mass than non-seam bowlers. The anthropometrics for seam bowlers within this study were similar to professional female seam bowlers reported within the literature [8, 22]. It is anticipated that seam bowlers will often be taller than other playing positions, as having a higher point of release when bowling is thought to be advantageous to bowling performance [6]. A higher ball release point will produce greater bounce from the pitch, making it more difficult for batters to execute their shot. It has also been recently reported that professional female seam bowlers with a higher ball release speed had a greater body mass than slower professional female seam bowlers (71.27 ± 8.49 kg vs. 65.95 ± 4.22 kg), which may partially explain why seam bowlers reported higher body mass than other playing positions [8]. Greater mass might be advantageous, by increasing the available whole-body linear momentum from the run-up to be transferred into angular momentum about the front foot during the delivery stride, leading to great ball release speeds [23]. In addition, the greater stature of seam bowlers may also contribute to the higher body mass reported for seam bowlers within the current investigation. Interestingly, despite the greater body mass reported for seam bowlers in comparison to non-seam bowlers, there was no difference in sum-of-8 skinfold measurements between positions among professional cricketers. This indicates a similar body composition between playing positions, despite the different requirements of the various positions within cricket match-play. Overall, the anthropometric variables presented add to the limited scientific research describing the physical characteristics of female cricketers.

Lower-limb power in both the vertical and horizontal planes has been suggested to be important to cricket performance, irrespective of playing position, as explosive lower-limb power would be anticipated to aid in fielding, power-hitting and seam bowling [5, 24, 25]. Despite the proposed importance of lower-limb power for cricket performance, there was no

differences in playing standard for either the broad jump or CMJ tests. These results may look to highlight the skill-based nature of cricket match-play and selection, with perhaps other skill-based factors deemed more important or necessary for competing at a professional level within female cricket. Interestingly, non-seam bowlers possessed great lower-limb horizontal power than seam bowlers, as measured by the broad jump. This result is in accordance with previous research in international male cricketers from an emerging cricket nation, reporting that batters had superior broad jump performance than seam bowlers [26]. However, it is important to note that the current study combined all non-seam bowling playing positions (i.e., batters, spin bowlers, wicketkeepers).

The IMTP measure of lower-limb maximal isometric strength presented a difference between Seam bowlers and non-seam bowlers for peak force. Professional female cricketers produced a significantly higher absolute peak vertical force when compared to non-professional female cricketers. This finding could be linked to the *moderately*, non-significant, higher body mass recorded for professional cricketers in comparison to non-professional cricketers. It was anticipated that seam bowlers would possess higher levels of lower limb strength, as high force production capacity has been suggested as a pre-requisite for tolerating the high ground reaction forces experienced during the deliver stride, as well as to optimise ball release speed within male seam bowling scientific literature [24, 27]. However, the results of the current investigation reported that non-seam bowlers possessed greater lower-limb strength. The ~25% greater normalised to bodyweight peak vertical force reported from the IMTP for non-seam bowlers may suggest that female seam bowlers adopted different strategies to developing ball release speed than their male counterparts. This perspective is supported by Felton, Lister [23] who reported that professional female seam bowlers generate less whole-body linear momentum during their run-up than males, relying more on contributions from whole-body angular momentum and large rotator muscles during the delivery stride to maximise ball release speed. These changes would put less of a reliance upon lower-limb strength for the generation of ball release speed. Furthermore, the values recorded within the current study for professional seam bowlers ($2.68 \pm 1.15$ N Kg$^{-1}$) for the IMTP are ~32% less than that recorded for professional female seam bowlers within the literature [8].

The run-2 test is a cricket specific measure of speed, as players are required to sprint between the wickets, with a bat, as they would during match-play. The results of this study indicated that professional cricketers were faster at performing the run-2 test than non-professional cricketers when turning on their dominant foot. Furthermore, professional non-seam bowlers demonstrated a *moderate*, non-significant trend of faster run-2 performance in comparison to professional seam bowlers and both non-seam bowlers and non-professional seam bowlers. The inclusion of batters, who are often required to produce several cricket specific sprints when running between the wickets during match-play, within the professional non-seam bowlers' group may partially explain this trend.

Professional cricketers reported a *moderately* higher Yo-Yo-IR1 scores than non-professional cricketers, while non-seam bowlers produced a *larger* run distance than seam bowlers. Professional non-seam bowlers produced the highest levels of aerobic endurance ($1234 \pm 321$ m) among all groups. Interestingly, this is less than that reported within the literature for other field-based, female athletes. For example, Division I collegiate women soccer players recorded $1666 \pm 473$ m for the Yo-Yo-IR1 [21]. The lower distance run for female cricketers within this investigation may not only be a result of the different sporting demands, but a consequence of domestic female cricket only recently becoming professional, limiting the exposure of female athletes to structured, physical preparation and coaching. This is of particular interest as recent research has identified that professional male cricketers, who have had greater exposure to structured, physical preparation practices and coaching, reported similar Yo-Yo-IR1 scores to

professional male soccer players (professional male cricketers = ~2426 m vs. top-elite male footballers = ~ 2302 m) [12]. Nonetheless, the greater aerobic endurance of non-seam bowlers in comparison to seam bowlers found in this study was surprising. Match demand data within men's cricket demonstrates that seam bowlers are required to cover greater distances, often at higher intensities than all other playing positions [2]. This greater physiological demand placed upon seam bowlers would likely necessitate increased aerobic endurance. However, Weldon, Clarke [3], also recently reported only a *small* difference in Yo-Yo-IR1 performance between seam bowlers and batters among international male cricketers from an emerging cricket nation. Perhaps, the test design of the Yo-Yo-IR1 which requires 180-degree turns favours batters, who would be anticipated to be more efficient in this movement pattern due to the frequency of performing 180-degree turns when running between the wickets. This could also explain why non-seam bowlers outperformed seam bowlers for the Yo-Yo-IR1 test within the current investigation, although to date, no research has reported on the match demands of female cricket.

There was no difference in linear sprint performance over both 0–10 m and 0–20 m were found between playing standard and position with only *trivial* to *small* effect sizes present. The times recorded for linear speed within the current investigation are similar to elite women's rugby union players [20] and faster than recreationally active female university students [28]. The lack of difference for playing standard in sprint performance, may again, highlight the skill-based nature of player selection and competition for female cricket, as well as linear sprint performance being of importance, regardless of positional role within the team.

There are certain limitations to this study. The non-seam bowler categorisation was unable to be further divided into specific playing positions (i.e., batters, spin bowlers, wicketkeepers), due to sample size constraints. Future research should look to assess the physical profile of all specific positions within female cricket. Researchers should seek to investigate the links between lower-limb power and cricketing performance across each playing position for female cricketers, to allow for greater insights into the links between lower-limb power and position specific cricket performance. Additionally, the data collection period was during the off-season phase of competition, which may limit how the presented physical profiles may relate to different phases of competition when structured physical training and match-play is implemented. There is a need for future research to assess the physical demands of match-play for female cricketers to better understanding the optimal physical profile for match-play. Clearly there is a need for further research into the female cricketer, as there is currently a lack of consistency and understanding regarding some aspects of their physical profile. For example, additional research investigating the relationship between physical profiles and years of cricketing experience would be worthwhile. Nonetheless, this is the first study to present the physical profile of female cricketers across different playing standards and positions and provides valuable information for S&C practitioners and coaches regarding the physical standards of female cricketers.

## Conclusion

The playing standard and positional physical profiles reported within the current investigation can be used for S&C practitioners and coaches across women's professional and non-professional cricket. The current results present normative data for professional and non-professional female cricketers across position types which can be used for comparison and to inform training requirements by practitioners via a gap analysis. Specifically, individual testing results from female cricketers can be compared to the current data set to highlight physical capacities

which may need to be developed to either enhance their position specific role within the team or bridge the game between competition playing standard.

## Supporting information

**S1 Table. Individual playing and anthropometric testing results.** Cm = centimetres; kg = kilograms; mm = millimetres; n/a = non-applicable.
(DOCX)

**S2 Table. Individual best result for each participant across physical capacity testing.**
m = metres; s = seconds; CMJ = countermovement jump; IMTP = isometric mid-thigh pull; cm = centimetres; N = newtons; BW = bodyweight.
(DOCX)

## Author Contributions

**Investigation:** Thomas A. Brazier.

**Methodology:** Thomas A. Brazier, Jamie Tallent, Samuel J. Callaghan.

**Project administration:** Thomas A. Brazier, Samuel J. Callaghan.

**Resources:** Thomas A. Brazier, Samuel J. Callaghan.

**Software:** Thomas A. Brazier.

**Supervision:** Jamie Tallent, Stephen D. Patterson, Samuel J. Callaghan.

**Writing – original draft:** Thomas A. Brazier, Jamie Tallent, Samuel J. Callaghan.

**Writing – review & editing:** Thomas A. Brazier, Jamie Tallent, Stephen D. Patterson, Louis P. Howe, Samuel J. Callaghan.

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
