## [Decision Letter · Decision Letter 0]

28 Sep 2023

PONE-D-23-22136The physical profile of female cricketers: An investigation between playing standard and positionPLOS ONE

Dear Dr. Brazier,

Thank you for submitting your manuscript to PLOS ONE. After careful consideration, we feel that it has merit but does not fully meet PLOS ONE’s publication criteria as it currently stands. Therefore, we invite you to submit a revised version of the manuscript that addresses the points raised during the review process.

 Dear Authors. two experts in the field reviewed your manuscript reporting several issues you should consider during the revision process. Please submit your revised manuscript by Nov 12 2023 11:59PM. If you will need more time than this to complete your revisions, please reply to this message or contact the journal office at plosone@plos.org. Please include the following items when submitting your revised manuscript:A rebuttal letter that responds to each point raised by the academic editor and reviewer(s). You should upload this letter as a separate file labeled 'Response to Reviewers'.A marked-up copy of your manuscript that highlights changes made to the original version. You should upload this as a separate file labeled 'Revised Manuscript with Track Changes'.An unmarked version of your revised paper without tracked changes. You should upload this as a separate file labeled 'Manuscript'.

We look forward to receiving your revised manuscript.

Kind regards,

Emiliano Cè

Academic Editor

PLOS ONE

Journal Requirements:

"No". 

"No".

Reviewers' comments:

Reviewer's Responses to Questions

**Comments to the Author**

1. Is the manuscript technically sound, and do the data support the conclusions?

Reviewer #1: Yes

Reviewer #2: Partly

2. Has the statistical analysis been performed appropriately and rigorously? 

Reviewer #1: Yes

Reviewer #2: Yes

3. Have the authors made all data underlying the findings in their manuscript fully available?

Reviewer #1: Yes

Reviewer #2: Yes

4. Is the manuscript presented in an intelligible fashion and written in standard English?

Reviewer #1: Yes

Reviewer #2: Yes

5. Review Comments to the Author

Reviewer #1: Title: The physical profile of female cricketers: An investigation between playing standard and position

First of all, the reviewer would like to thank the authors for their work and efforts in trying to improve sports science knowledge. The study emphasizes the need to understand the physical profile of female cricketers according to playing standard and position. Overall, the study is well designed and well-written, with a great introduction proposing the usefulness of the topic. I suggest that the manuscript can be accepted without any revision.

Reviewer #2: In the introduction, it is stated that the aim is to characterize the field positions, but in the methods, only seam bowlers are reported, without specifying what is meant by "non-seam bowlers." I recommend clarifying this aspect and maintaining consistency throughout the paper. In fact, it is only in the discussion section that it becomes clear what is meant by non-seam bowlers.

Please, add that the research was according to the declaration of Helsinki

Did you calculate the optimal sample size? Why were 45 participants selected?

Regarding the exclusion criteria, how and whether past/history of injuries were considered?

Did you request participants to abstain from caffeine and adrenergic substances before performance? If not, did you monitor this? Previous studies have shown that caffeine affects jump and sprint performance.

Were the tests conducted at the same time of day? Performance could be influenced by circadian rhythms.

Did you inquire about how many years the participants had been playing cricket? Additionally, what kind of training differentiates professionals from non-professionals? Moreover, do different roles train differently, not only in terms of technique but also in terms of physical training?

What about the angle during the Isometric Mid-Thigh Pull? Was it standardized or measured?

Why did you choose the Isometric Mid-Thigh Pull as the test to measure lower-limb maximal isometric strength? It is a test that requires high familiarity. In connection with this, was a familiarization session planned for each test?

I suggest providing a more detailed explanation of the purpose of using skinfold measurements.

Change "2X2 ANOVA" to "two-way ANOVA."

I recommend presenting the results in the text along with their respective percentages and effect sizes in parentheses for reason of readability.

Why do the 49 participants in Table 1 not correspond to the 45 participants mentioned in the methods section?

Figure 1: How could you explain the differences within the same groups, especially among the professionals? What about the outliers?

In the discussion section, at the end of each paragraph, the authors mention the limitations of the current study and express the need for further investigation. This highlights the study's limitations and weaknesses. Therefore, I would recommend grouping the limitations and future prospective only at the end of the discussion paragraph.

6. PLOS authors have the option to publish the peer review history of their article (what does this mean?). If published, this will include your full peer review and any attached files.

Reviewer #1: No

Reviewer #2: No

---

## [Author Response · Author response to Decision Letter 0]

4 Mar 2024

Responses to Reviewer 1 and 2: We thank the reviewers for their time, constructive comments, and critique of our manuscript and appreciate the positive comments that we have received. We have addressed each point and as a result we feel the manuscript has been much improved. The comments and subsequent responses are addressed in a systematic fashion and are summarised below.

PONE-D-23-22136

The physical profile of female cricketers: An investigation between playing standard and position PLOS ONE 

Reviewers Comments

Reviewer #1: First of all, the reviewer would like to thank the authors for their work and efforts in trying to improve sports science knowledge. The study emphasizes the need to understand the physical profile of female cricketers according to playing standard and position. Overall, the study is well designed and well-written, with a great introduction proposing the usefulness of the topic. I suggest that the manuscript can be accepted without any revision. 

Response: The authors would like to thank the reviewer for their time in reviewing the manuscripts and their positive comments regarding the relevance and impact of the research.

Reviewer #2: 

Comment: In the introduction, it is stated that the aim is to characterize the field positions, but in the methods, only seam bowlers are reported, without specifying what is meant by "non-seam bowlers." I recommend clarifying this aspect and maintaining consistency throughout the paper. In fact, it is only in the discussion section that it becomes clear what is meant by non-seam bowlers. 

Response: Firstly, the authors would like to thank the reviewer for their time and comments regarding the submitted manuscript, it is greatly appreciated. 

Many thanks for raising this, I can now see how further clarification is needed throughout the manuscript. The description within the methods has been amended to reflect this. I hope this now gives readers of the manuscript further understanding of playing positions.

“The positional classification was defined by the head coach of each respective team, with a seam bowler defined as a player whose primary role within the team was seam bowling. Non-seam bowlers were classified as cricketers that occupied all other positions (i.e., spin bowlers, batters, and wicket keepers).”

Comment: Please, add that the research was according to the declaration of Helsinki

Response: This has now been added to the manuscript.

“The research was approved by the institutional ethics committee, in agreement with the declaration of Helsinki”.

Comment: Did you calculate the optimal sample size? Why were 45 participants selected?

Response: The authors agree that looking to undertake power calculations to determine the optimal sample size for a study is often a valuable process when conducting research. However, for the current study we do not believe that undertaking sample size calculations is recommended for several reasons. Firstly, due to the novel nature of the research, as there is currently no physical capacity data available for professional or non-professional English-based female cricketers, the authors feel that the use of any available effect sizes within the research field would not be appropriate – setting an expected effect size difference based upon previous research is a key step when performing power calculations to determine the sample size of a study. Furthermore, recent research from Woodhouse et al., (2022) has demonstrated significant changes in the physical capacity of female athletes over time. Consequently, any previous research related to the physical capacity of female athletes may not be appropriate to use as a suggested effect size for power analysis calculations. Therefore, the authors made the decision to recruit as many professional and non-professional female cricketers within English domestic cricket as possible. It is worth noting that the recruited 54 participants represented approximately 18% of the entire population of English domestic cricket professional and non-professional players which the authors believe allows for valuable translation of the studies finding to the wider population.

Woodhouse, L.N., Tallent, J., Patterson, S.,D and Waldron, M. (2022). International female rugby union players anthropometric and physical performance characteristics: A five-year longitudinal analysis by individual playing positions. Journal of Sport Sciences, 40(4), 370-378.

Comment: Regarding the exclusion criteria, how and whether past/history of injuries were considered?

Response: A physical activity readiness questionnaire detailing past injuries was considered by the principal investigator prior to participation of testing. Participant’s that were free from current injury took part in all testing. Participants who had injuries could only complete certain test and these were decided in collaboration with the team’s physiotherapist. Additional text has been added to provide further clarity regarding the injury contra-indicators for participation and how this was determined. 

“…….as determined by a physical activity readiness questionnaire screening and discussion and clearance with each domestic teams, qualified physiotherapists.”

Comment: Did you request participants to abstain from caffeine and adrenergic substances before performance? If not, did you monitor this? Previous studies have shown that caffeine affects jump and sprint performance.

Response: Additional text has been added regarding the use of caffeine and adrenergic substances prior to testing.

“All Participants were instructed to refrain from vigorous physical exercise, caffeine or any androgenic substance in the 24-hours prior to data collection, and to maintain their normal dietary habits.”

Comment: Were the tests conducted at the same time of day? Performance could be influenced by circadian rhythms.

Response: Additional text has been added regarding the standardisation of the time of testing for all participants.

“Testing was conducted in the indoor training facilities of each of the six domestic regional teams included within the study, at the same time of day.”

Comment: Did you inquire about how many years the participants had been playing cricket? Additionally, what kind of training differentiates professionals from non-professionals? Moreover, do different roles train differently, not only in terms of technique but also in terms of physical training?

Response: The number of years by which participants had been participating in cricket was not recorded as part of the data collection process. The focus of the study was on whether physical capacity differed between playing standards, not on more global what factors which differentiated between playing standards. Nonetheless, additional text has been added to the limitations paragraph discussing this element. 

“Clearly there is a need for further research into the female cricketer, as there is currently a lack of consistency and understanding regarding some aspects of their physical profile. For example, additional research investigating the relationship between physical profiles and years of cricketing experience would be worthwhile.”

Regarding the exposure to training at the different competition levels, both professional and non-professional pathway athletes will be exposed to structured and supervised strength and conditioning. Players across both standards and position types will receive individualised training programmes based upon the needs of the sport, which will include elements of position specific training, but also individual capacity/physical development, which will influence the programme provided to the athlete. 

Comment: What about the angle during the Isometric Mid-Thigh Pull? Was it standardized or measured?

Response: A self-selected posture, within set knee and hip angle ranges, was adopted. This was based upon the research by Comfort et al., (2015) which found that there was no significant difference between specific hip and knee angles, and the athlete’s preferred position when performing the IMTP. Additional text has been added to the methods to provide further clarity regarding the IMTP procedures. 

“This positioning allowed participants to maintain their preferred position between a knee angle range of 120°-150°, and a hip angle range of 125°-145° during testing, which has been shown within previous research to result in no statistical difference in peak force variables when performing the IMTP (Comfort et al., 2015).”

Comfort, P., Jones, P.A., McMahon, J.J. and Newton, R. (2015). Effect of knee and trunk angle on kinetic variables during the isometric midthigh pull: Test-retest reliability. International Journal of Sports Physiology and Performance, 10(1), 58-63.

Comment: Why did you choose the Isometric Mid-Thigh Pull as the test to measure lower-limb maximal isometric strength? It is a test that requires high familiarity. In connection with this, was a familiarization session planned for each test?

Response: The IMTP was chosen as an assessment of lower-limb strength as it is currently part of the physical testing battering required for national level cricketers within an international test playing nation (Callaghan et al., 2021; Callaghan et al., 2023). Anecdotally, the IMTP is also widely utilised within County Cricket within the UK as well, and all participants had previous exposure to this test. Additional text has been added to the manuscript to provide further clarity regarding the procedures of the IMTP. 

“All participants were familiar with the IMTP testing procedures.”

Callaghan, S. J., Lockie, R. G., Tallent, J., Chipchase, R. F., Andrews, W. A., & Nimphius, S. (2023). The effects of strength training upon front foot contact ground reaction forces and ball release speed among high-level cricket pace bowlers. Sports Biomechanics, 1-17.

Callaghan, S. J., Govus, A. D., Lockie, R. G., Middleton, K. J., & Nimphius, S. (2021). Not as simple as it seems: Front foot contact kinetics, muscle function and ball release speed in cricket pace bowlers. Journal of sports sciences, 39(16), 1807-1815.

Comment: I suggest providing a more detailed explanation of the purpose of using skinfold measurements.

Response: The purpose of the current research project was to investigate the physical differences between playing positions and standards. While it was not possible to investigate skinfold measurements between playing standards, the authors believe exploring differences in skinfold measurements between playing positions for professional female crickets, aligns with the purpose of the research project. Furthermore, the authors feel that the skinfold measurement provides valuable and novel information to the scientific community about the physical profile of elite female cricketers. It should also be noted that the skinfold measurement is part of the National Cricketing Bodies player profiling requirements. Consequently, additional text has been added to the manuscript to justify the inclusion of this metric. 

“Skinfold assessment has previously been shown to link to physical capacity metrics and also forms part of the testing procedures of the English and Wales Cricket Board.”

Comment: Change "2X2 ANOVA" to "two-way ANOVA."

Response: This has been amended.

“assessed with a two-way ANOVA”

Comment: I recommend presenting the results in the text along with their respective percentages and effect sizes in parentheses for reason of readability.

Response: Thank you for the recommendation regarding the presentation of results. However, the authors do believe that this would result in information presented in tables and/or figures being repeated in text, which should look to be avoided. Hence, the authors believe that it is appropriate to leave the presenting of results within text as originally submitted. 

Comments: Why do the 49 participants in Table 1 not correspond to the 45 participants mentioned in the methods section?

Response: The discrepancy between the number of participants outlined in the methods section (i.e., 54 participants), and that presented in Table 1, is because of not all participants being able to complete each test. Ideally, all participants would have completed each test, but this was not always possible, due to several reasons. For transparency, the number of participants for each test is presented in Table 1. 

Comment: Figure 1: How could you explain the differences within the same groups, especially among the professionals? What about the outliers?

Response: The results of the current study reported no significant interaction affect between playing standard and position. Hence, while the figure/s will show visual differences within the same group, this did not reach statistical significance. Therefore, the authors have chosen to focus the discussion of results around the significant, main effect differences found. The discussion of outlier participants within the data set, is beyond the scope of the current investigation, and does not align with the goal of looking for more global differences between playing standard and positions. 

Comment: In the discussion section, at the end of each paragraph, the authors mention the limitations of the current study and express the need for further investigation. This highlights the study's limitations and weaknesses. Therefore, I would recommend grouping the limitations and future prospective only at the end of the discussion paragraph. 

Response: Thanks for your comment, recommendations from individual paragraphs have been removed and added to a single limitations paragraph within the discussion. 

There are certain limitations to this study. The non-seam bowler categorisation was unable to be further divided into specific playing positions (i.e., batters, spin bowlers, wicketkeepers), due to sample size constraints. Future research should look to assess the physical profile of all specific positions within female cricket. Researchers should seek to investigate the links between lower-limb power and cricketing performance across each playing position for female cricketers, to allow for greater insights into the links between lower-limb power and position specific cricket performance. Additionally, the data collection period was during the off-season phase of competition, which may limit how the presented physical profiles may relate to different phases of competition when structured physical training and match-play is implemented. There is a need for future research to assess the physical demands of match-play for female cricketers to better understanding the optimal physical profile for match-play. Clearly there is a need for further research into the female cricketer, as there is currently a lack of consistency and understanding regarding some aspects of their physical profile. For example, additional research investigating the relationship between physical profiles and years of cricketing experience would be worthwhile. Nonetheless, this is the first study to present the physical profile of female cricketers across different playing standards and positions and provides valuable information for S&C practitioners and coaches regarding the physical standards of female cricketers.

---

## [Decision Letter · Decision Letter 1]

25 Mar 2024

PONE-D-23-22136R1The physical profile of female cricketers: An investigation between playing standard and positionPLOS ONE

Dear Dr. Callaghan,

Thank you for submitting your manuscript to PLOS ONE. After careful consideration, we feel that it has merit but does not fully meet PLOS ONE’s publication criteria as it currently stands. Therefore, we invite you to submit a revised version of the manuscript that addresses the points raised during the review process.

**ACADEMIC EDITOR: **Dear Authors, your manuscript version 2 has been reviewed by one expert in the field that retrieved some minor issues you should consider during the revision process.============================

We look forward to receiving your revised manuscript.

Kind regards,

Emiliano Cè

Academic Editor

PLOS ONE

Journal Requirements:

Reviewers' comments:

Reviewer's Responses to Questions

**Comments to the Author**

1. If the authors have adequately addressed your comments raised in a previous round of review and you feel that this manuscript is now acceptable for publication, you may indicate that here to bypass the “Comments to the Author” section, enter your conflict of interest statement in the “Confidential to Editor” section, and submit your "Accept" recommendation.

Reviewer #2: All comments have been addressed

2. Is the manuscript technically sound, and do the data support the conclusions?

Reviewer #2: Yes

3. Has the statistical analysis been performed appropriately and rigorously? 

Reviewer #2: No

4. Have the authors made all data underlying the findings in their manuscript fully available?

Reviewer #2: Yes

5. Is the manuscript presented in an intelligible fashion and written in standard English?

Reviewer #2: Yes

6. Review Comments to the Author

Reviewer #2: Some comments have been addressed and the paper seems to me more clear; however, some others require further attention.

Which version of the Declaration of Helsinki were the authors referring to?

Did the authors intend to refer to adrenergic or androgenic substances?

Regarding the statistical details, I recommend including supplementary materials if the authors prefer tables or figures instead of textual descriptions. It is particularly crucial to comprehend the magnitude of these differences, especially considering that the authors opted not to provide the sample size calculation.

7. PLOS authors have the option to publish the peer review history of their article (what does this mean?). If published, this will include your full peer review and any attached files.

Reviewer #2: No

---

## [Author Response · Author response to Decision Letter 1]

3 Apr 2024

Reviewer 2 Comments

Reviewer #2: Some comments have been addressed and the paper seems to me more clear; however, some others require further attention.

The authors would like to thank the reviewer for their time and consideration in reviewing the manuscript. Please see below specific response to the comments/questions raised. 

Which version of the Declaration of Helsinki were the authors referring to?

Thank you for your comment. Amendments have been made to the manuscript.

“The research was approved by the institutional ethics committee, in agreement with the declaration of Helsinki, version seven.”

Did the authors intend to refer to adrenergic or androgenic substances?

Thank you for your comment. Amendments have been made to the manuscript.

“All Participants were instructed to refrain from vigorous physical exercise, caffeine, or any adrenergic substance in the 24-hours prior to data collection, and to maintain their normal dietary habits.”

Regarding the statistical details, I recommend including supplementary materials if the authors prefer tables or figures instead of textual descriptions. It is particularly crucial to comprehend the magnitude of these differences, especially considering that the authors opted not to provide the sample size calculation.

Thank you for your recommendation. As noted, the statistical details (i.e., F statistic, degrees of freedom and p-value) are presented in text within the results section. The authors believe that due to the concise results section, readers can easily access and interpret these statistical details from the text. Hence, the authors do not believe that this information needs to be transferred to a table, and feel that the current presentation of statistical details is appropriate. However, if the reviewer is adamant that this change is necessary, and the editor agrees, the authors will consider this amendment to the manuscript.

---

## [Decision Letter · Decision Letter 2]

10 Apr 2024

The physical profile of female cricketers: An investigation between playing standard and position

PONE-D-23-22136R2

Dear Dr. Callaghan,

We’re pleased to inform you that your manuscript has been judged scientifically suitable for publication and will be formally accepted for publication once it meets all outstanding technical requirements.

Kind regards,

Emiliano Cè

Academic Editor

PLOS ONE

Additional Editor Comments (optional):

Reviewers' comments:

Reviewer's Responses to Questions

**Comments to the Author**

1. If the authors have adequately addressed your comments raised in a previous round of review and you feel that this manuscript is now acceptable for publication, you may indicate that here to bypass the “Comments to the Author” section, enter your conflict of interest statement in the “Confidential to Editor” section, and submit your "Accept" recommendation.

Reviewer #2: All comments have been addressed

2. Is the manuscript technically sound, and do the data support the conclusions?

Reviewer #2: (No Response)

3. Has the statistical analysis been performed appropriately and rigorously? 

Reviewer #2: Yes

4. Have the authors made all data underlying the findings in their manuscript fully available?

Reviewer #2: Yes

5. Is the manuscript presented in an intelligible fashion and written in standard English?

Reviewer #2: Yes

6. Review Comments to the Author

Reviewer #2: The comments are addressed and the paper is now more clear and precise. I would suggest to replace version 7 of the Declaration of Helsinki with "latest version of Declaration of Helsinki".

7. PLOS authors have the option to publish the peer review history of their article (what does this mean?). If published, this will include your full peer review and any attached files.

Reviewer #2: No

---

## [Editor Report · Acceptance letter]

8 May 2024

PONE-D-23-22136R2 

PLOS ONE

Dear Dr. Brazier, 

I'm pleased to inform you that your manuscript has been deemed suitable for publication in PLOS ONE. Congratulations! Your manuscript is now being handed over to our production team.

Kind regards, 

on behalf of

Prof. Emiliano Cè 

Academic Editor

PLOS ONE